# Cognitively unimpaired adults' reactions to disclosure of amyloid PET scan results

**Emily A. Largent**[1]*, **Kristin Harkins**[2], **Christopher H. van Dyck**[3], **Sara Hachey**[4],
**Pamela Sankar**[1], **Jason Karlawish**[1,2,5]

**1** Department of Medical Ethics and Health Policy, University of Pennsylvania Perelman School of Medicine, Philadelphia, Pennsylvania, United States of America, **2** Department of Medicine, University of Pennsylvania Perelman School of Medicine, Philadelphia, Pennsylvania, United States of America, **3** Yale University School of Medicine, New Haven, Connecticut, United States of America, **4** Lewis Katz School of Medicine at Temple University, Philadelphia, Pennsylvania, United States of America, **5** Department of Medicine, University of Pennsylvania Perelman School of Medicine, Philadelphia, Pennsylvania, United States of America

* elargent@pennmedicine.upenn.edu

**Data Availability Statement:** Data (i.e., interview transcripts) cannot be shared publicly because of the potential for identification of participants. The Penn IRB has reviewed this material with us and

## Abstract

### Importance

Clinical guidelines currently recommend against amyloid imaging for cognitively unimpaired persons. The goal of Alzheimer's disease (AD) prevention, together with advances in understanding the pathophysiology of AD, however, has led to trials testing drugs in cognitively unimpaired persons who show evidence of AD biomarkers. Assuming the eventual success of such trials, millions of patients will be affected. There is a need to understand the effects of biomarker disclosure on those individuals.

### Design

The Study of Knowledge and Reactions to Amyloid Testing (SOKRATES) involved 2 semi-structured telephone interviews with individuals who received amyloid PET scan results as part of screening for research participation. Post-disclosure interviews were conducted at 4 to 12 weeks and again 1 year later. Data were collected from November 5, 2014 to November 30, 2016. Interviews were transcribed and coded in NVivo 12.0.

### Participants

80 adults aged 65 and older: 50 who received "elevated" and 30 who received "not-elevated" amyloid PET scan results.

### Main outcomes

Interviews examined four domains: (1) comprehension of the amyloid PET scan result; (2) implications of the result for sense of self, memory, and future; (3) sharing of results with others; and (4) AD risk-reduction behaviors.

agrees that, given that participants were drawn from a particular Alzheimer's disease treatment study with strict enrolment criteria and that participants are discussing personal information at length (e.g., about work-retirement tradeoffs), the potential for re-identification remains high despite the fact that the transcripts have been stripped of research participant identifiers (e.g., names and places). Further, given the sensitive nature of biomarker results (e.g., implications for insurability), this concern about re-identification is heightened. Data access requests may be made to the Penn IRB at: IRB@pobox.upenn.edu.

**Funding:** This research was supported by the Investigator-Initiated Research Grant "Study of Knowledge and Reactions to Amyloid Testing" from the Alzheimer's Association awarded to Dr. Karlawish and by grant P30AG010124 from the National Institute on Aging (Dr. Karlawish, co-investigator). Dr. Largent is supported by NIA K01-AG064123. The funders had no role in the design and conduct of the study; collection, management, analysis, or interpretation of the data; preparation or approval of the manuscript; and decision to submit the manuscript for publication.

**Competing interests:** The authors have declared that no competing interests exist.

## Results

Participants who received an elevated amyloid PET scan result viewed the result as more serious and sensitive than other medical test results given its unique implications for identity, self-determination, and stigma. In contrast, participants who received a not-elevated amyloid PET scan result described feeling relief and reinterpreted perceived memory impairments as normal aging. Participants with elevated amyloid reported contemplating and making more changes to health behaviors and future plans than their peers with not-elevated amyloid.

## Conclusions

Clinical practice in the diagnosis and treatment of persons with preclinical AD, a stage of the disease defined by the presence of biomarkers in the absence of cognitive impairment, will need to address matters of identity, stigma, and life-planning.

## Introduction

Alzheimer's disease (AD) is being reconceptualized as a disease that begins in a preclinical stage characterized by the presence of AD biomarkers in the absence of cognitive impairment.[1][2] An estimated 46.7 million Americans have preclinical AD, though not all will progress to the clinical stages of mild cognitive impairment (MCI) or dementia.[3] At present, diagnostic guidelines recommend against imaging for the diagnosis of AD in cognitively unimpaired persons.[4,5] Should the preclinical AD construct be validated and become a target of therapeutic intervention, however, cognitively unimpaired older adults will likely be routinely screened for AD biomarkers.

Looking to the future, clinicians need to understand both how patients experience the emerging disease stage of preclinical AD and how best to talk about biomarker results in the pre- and post-test context. Anticipating these needs, the same studies designed to validate preclinical AD and to test novel interventions in the preclinical stage are an opportunity to understand the practical and ethical dimensions of biomarker disclosure to cognitively unimpaired persons.[6–8] Interest in learning biomarker results is high among at-risk individuals and is supported by investigators.[9–11] The extant empirical literature on disclosure of amyloid imaging results to cognitively unimpaired adults is, however, limited.[12,13] The more we understand their understanding of and reactions to biomarker disclosure, the better we can prepare for future clinical practice.[14]

Here, we report the results of the Study of Knowledge and Reactions to Amyloid Testing (SOKRATES), a longitudinal qualitative study of cognitively unimpaired adults ages 65 to 85 who learned the result of an amyloid positron emission tomography (PET) scan in order to enroll in a clinical trial.

## Methods

SOKRATES participants were recruited from the Anti-Amyloid Treatment in Asymptomatic Alzheimer's Study (A4) (NCT0200835)—a secondary prevention trial testing whether solanezumab can slow cognitive decline in persons with amyloid accumulation—and its companion longitudinal cohort study, Longitudinal Evaluation of Amyloid Risk and Neurodegeneration (LEARN) (NCT02488720).[15] A4 participants had evidence of amyloid plaque build-up (i.e., an "elevated" amyloid PET scan result), while LEARN participants screen-failed for A4 solely on the basis of not having amyloid accumulation (i.e., a "not-elevated" amyloid PET

scan result). Inclusion critera required that participants were cognitively unimpaired at baseline; they scored within normal limits on baseline cognitive testing and were rated a zero (no impairment) on the Clinical Dementia Rating (CDR) dementia-staging instrument.[16] Subjective cognitive complaints were permitted.

A4 and LEARN participants underwent a standardized amyloid disclosure process.[17] Participants were pre-screened for depression and anxiety, and completed an educational session with a comprehension check in which they received both verbal and written information about amyloid imaging, including possible results, their meaning, and their implications for risk of future cognitive decline. The study guide explains that elevated amyloid "does not necessarily mean you will develop AD-related memory loss" but can be associated with an increased risk; it further explains that "not elevated" does not mean you will never develop AD or "elevated amyloid" in the future. Amyloid imaging occurred on a separate day from the education session, and disclosure of the imaging results occurred on a separate day from imaging. Site investigators disclosed the amyloid PET scan results in-person using standardized talking points. Participants received a post-disclosure follow up phone call and regular monitoring of mood and well-being throughout the study.

For SOKRATES, 50 participants with "elevated" amyloid and 30 participants with "not-elevated" amyloid completed a semi-structured interview 4 to 12 weeks after disclosure of their amyloid PET scan results (T1); 47 and 30 of these individuals, respectively, completed a 12-month follow-up interview (T2). T1 interviews examined four domains: (1) comprehension of the amyloid result; (2) implications for sense of self, memory, and future; (3) sharing of results with others; and (4) risk-reduction behaviors. T2 interviews re-examined participants' reports from the initial interview.

All interviews occurred between November 5, 2014 and November 30, 2016 and were conducted by one interviewer (KH). Interviews were recorded, transcribed, and analyzed in NVivo qualitative analysis software, version 11.0 (QSR International). The methods have previously been reported.[18] Briefly, the research team reviewed all transcripts to develop a coding scheme to reflect the four aforementioned domains and to capture themes that emerged during coding and analysis. This iterative process involved multiple consensus meetings to resolve coding discrepancies, regular checks on agreement using the Cohen coefficient for inter-coder reliability, and adjustments to the codebook with an audit trail of coding rules and decisions made. To allow comparisons between participants who received an "elevated" result and those who received a "not-elevated" result, we report the frequency of codes as a fraction of each group's total members.

The University of Pennsylvania Institutional Review Board (IRB) approved this study. The IRB approved a waiver of documentation of consent. SOKRATES participants were mailed or emailed written information about the study; that information was reviewed with them over the phone. Only then were participants asked to give verbal consent to be interviewed. Participants received a $20 gift card (choice of CVS or Amazon) in exchange for their participation.

## Results

Table 1 reports participant demographics. Analyses included searching for differences among groups based on demographic variables; no notable differences in results were observed by age or gender.

### Understanding of amyloid PET scan results

Most participants who received an "elevated" amyloid PET scan result (31 of 50 [62%]) understood that the result described an increased but uncertain risk of developing AD dementia.[18]

**Table 1. Demographic characteristics of SOKRATES subjects by amyloid status at time of the initial interview[a,b].**

| Characteristic | "Elevated" Amyloid (n = 50)[c] | "Not-Elevated" Amyloid (n = 30) |
|---|---|---|
|  | N (%) | N (%) |
| **Sex** |  |  |
| Male | 25 (50%) | 13 (43%) |
| Female | 25 (50%) | 17 (57%) |
| **Age[d]** |  |  |
| 65–74 | 35 (70%) | 25 (83%) |
| ≥ 75 | 15 (30%) | 5 (17%) |
| **Race/Ethnicity** |  |  |
| Caucasian/non-Hispanic | 49 (98%) | 28 (93%) |
| Asian | 1 (2%) | 0 |
| Caucasian/Hispanic | 0 | 1 (3%) |
| Multiracial/Hispanic | 0 | 1 (3%) |
| **Education** |  |  |
| High School | 1 (2%) | 0 |
| Some College or College Degree | 19 (38%) | 11 (37%) |
| Post Graduate Education | 30 (60%) | 19 (63%) |
| **Family history of Alzheimer's disease** |  |  |
| Yes | 40 (80%) | 21 (70%) |
| No | 10 (20%) | 8 (27%) |
| Unknown[e] |  | 1 (3%) |
| **Marital Status** |  |  |
| Married/Living with Partner | 36 (72%) | 26 (83%) |
| Divorced/Separated | 8 (16%) | 2 (7%) |
| Widowed | 4 (8%) | 2 (7%) |
| Single | 2 (4%) | 1 (3%) |
| **Employment Status** |  |  |
| Retired | 31 (62%) | 20 (67%) |
| Part-Time | 14 (28%) | 7 (23%) |
| Full-Time | 5 (10%) | 3 (10%) |

[a]Participating sites (n = 10) provided between 0–12 participants with elevated amyloid and 0–12 participants with not elevated amyloid

[b]Distribution of demographic characteristics did not differ by amyloid status at p = 0.05 level, by Chi-Square or Fisher's exact test

[c]Three elevated amyloid participants completed interview 1 and could not be reached for follow up

[d]Oversampled for greater representation among subjects 65–74 years of age due to the potentially greater significance of an amyloid PET scan imaging result for younger individuals

[e]One participant was adopted and unable to provide information on family history

Some (20 of 50 [40%]) felt it was ambiguous, and many of these participants wanted quantitative information about the amount of amyloid detected.

Half of the participants who received a "not-elevated" result (16 of 30 [53%]) understood their result to mean they were at decreased risk of developing AD dementia. Some (9 [30%]) emphasized that the result meant they did not presently have AD but acknowledged "it doesn't mean that I won't get it in the future." Few (3 [10%]) described their result as ambiguous or expressed a desire for additional information.

## Effect on perceptions of memory

A third of participants with elevated brain amyloid (18 of 50 [36%]) reported feeling that their memory was impaired *prior* to the amyloid PET scan. For them, receiving an "elevated" result validated their memory-related concerns. Another third (16 [32%]) reported becoming more aware of and more worried about memory issues after learning their result. One explained, "I'm starting to question more whether these 'senior moments' are related to amyloid plaques." At T2, these participants' perceptions of memory were largely unchanged.

Receiving a "not-elevated" result relieved participants' memory-related anxiety. Roughly half of these individuals (16 of 30 [53%]) described re-interpreting memory lapses as normal aging. One explained that the result "made me think that any memory problems I was having was just normal age related rather than . . . Alzheimer's." At T2, most participants with not-elevated brain amyloid continued to describe minor memory lapses as "normal aging," though several expressed frustration because they lacked an explanation for perceived memory issues. One explained, "[K]nowing that I don't have any amyloids, I'm saying, 'Well, what can it be?´´´

At T1, a notable minority of all participants (elevated brain amyloid: 16 of 50 [32%]; not-elevated brain amyloid: 12 of 30 [40%]) stated both that they had no concerns about their memory prior to the amyloid PET scan and that learning their result had no effect on their perceptions of memory. By T2, however, several participants with elevated brain amyloid expressed increased concern about their memory, while a few participants who had received a "not-elevated" imaging result noted memory issues but expressed "peace of mind" that these were not due to brain amyloid.

## Comparing amyloid PET scan results to other medical test results

Most participants (elevated brain amyloid: 33 of 50 [66%]; not-elevated brain amyloid: 23 of 30 [77%]) reported that the amyloid PET scan result was unlike results from other medical tests.

Half of the participants with elevated brain amyloid who felt the amyloid imaging result was different than other medical tests (16 of 33 [48%]) described it using words such as *sensitive*, *touchy*, *severe*, and *devastating*. Several explained that the amyloid PET scan result had unique implications for their sense of self, stating for example that "a colonoscopy isn't going to change who I am . . . this is my brain involved" or "[the result] speaks to who I am . . . my brain is a very critical part of me." A quarter of participants with elevated brain amyloid who felt the amyloid imaging result was different than other medical tests (9 of 33 [27%]) expressed concern that the result could have social consequences because "[l]osing your mental faculties is regarded by people differently than eye sight or hearing or anything else because they are seeing that you're less of a person" and "Alzheimer's has a negative stigma to it." Reasons for not sharing an "elevated" result with others mirrored these concerns and included desires "not to distress anyone," to respect others' desires not to know, to avoid negative social consequences such as being "shut out" or treated differently, and to prevent discrimination in employment, healthcare, or insurance.

Nearly half of participants who received a "not-elevated" result and reported that the amyloid PET scan result was different than other medical test results (11 of 23 [48%]) described it as a "research finding" rather than a clinical result.

## Feelings about the future

Participants with elevated brain amyloid expressed diverse feelings about the future (Table 2). They variously described feeling that their future was bleak (12 of 50 [24%]), unknown (27

**Table 2. Responses to the questions "How do you feel about your future?" and "Did you feel the same way before you learned your amyloid PET scan result?/ How did learning your result change how you feel about your future?"[a].**

| | "Elevated" Amyloid | | Representative Quote | "Not-Elevated" Amyloid | | Representative Quote |
|---|---|---|---|---|---|---|
| | Time 1 (N = 50) | Time 2 (N = 47) | | Time 1 (N = 30) | Time 2 (N = 30) | |
| **Bright Future** | 14 (28%) | 12 (25%) | "I'm convinced they'll find a cure for Alzheimer's, so I feel relieved that I have an advantage, that knowing I have it, I can be doing lifestyle things to help myself, and maybe really keep an eagle eye out for developments in the clinical world." | 19 (63%) | 17 (57%) | "Now that the probability is lower that I'm going to have Alzheimer's that gives the longer term forecast a lot more positive look to it." |
| **Bleak Future** | 12 (24%) | 6 (13%) | "Well, I'm less optimistic than I used to be about what my future would be like. I know over the years, several people who have had Alzheimer's. . . . They were more like vegetables. I don't look forward to that." | 2 (7%) | 1 (3%) | "I feel like I'm almost 75 and that's sort of the way it's going to be. It's not going to get better." |
| **Future Unknown** | 27 (54%) | 1 4 (30%) | "Well, I would say there are just a lot of question marks. . . just from the standpoint of not knowing whether I'm going to stay the same, get worse, and how soon that would happen. . . I have no information to guide me other than I had this elevated amyloids, it's not that much information to go on. . ." | 6 (20%) | 3 (10%) | "How do I feel about the future? It's an unknown. It's a big question mark, I guess." |
| **Not Thinking About Future** | 22 (44%) | 21 (45%) | "I don't think that far ahead. My future right now is. . . I'm kind of in the present or trying to be. I'm not worried. I can't be thinking that, "You know, okay, I'm 66. When I'm 86, I'm going to be this or that or the other thing." I'm busy trying to enjoy being 66." | 12 (40%) | 13 (43%) | "A lot of issues of the future are not things I really want to think about.. . . I'm just going with the flow, as it were." |

[a]Codes are not mutually exclusive. Responses were coded for whether each sentiment was expressed.

[54%]), or bright (14 [28%]). Reasons for feeling the future was bright included being optimistic by nature, hoping to "be one of the lucky ones" who escapes dementia, having faith in medical advances, and having family members who were healthy into their old age. No participants expressed relief after receiving an "elevated" result.

Participants who received a "not-elevated" amyloid PET scan result had more uniform feelings about the future. Two-thirds (19 of 30 [63%]) reported their future was bright, and two-thirds (19 [63%]) reported feeling relief. Nearly half [13 [43%]] reported both sentiments. One participant explained that he had been "living under this cloud that someday [AD] may get me. . . . [The amyloid PET scan result] took a lot off of my mind."

Nearly half of participants with elevated amyloid (T1: 22 of 50 [44%], T2: 21 of 47 [45%]) and also with not-elevated amyloid (T1: 12 of 30 [40%], T2: 13 of 30 [43%]) were present-focused. When asked directly about their future, some persisted in talking only about the present, while others described deliberately not thinking about the future.

## Health behaviors

Across T1 and T2, participants who received "elevated" and "not-elevated" results reported changing health behaviors, though change was more common among participants with elevated brain amyloid. Common health behavior changes are reported in Table 3. Many participants who had received an "elevated" result reported undertaking health behavior changes specifically to remain cognitively healthy and to delay or prevent cognitive symptoms. Representative explanations included: "I'm trying to eat more berries and nuts 'cause that can be healthy for the brain" or "I started running again . . . I read that that helps, maybe, possibly,

**Table 3. Health behavior: Material derived from responses to the questions: "Have you made any changes in your daily life based on knowing your amyloid scan result?/ Have you started/stopped doing anything?/Are you spending time differently?" As well as queries about specific health behaviors: Diet, exercise, medications/vitamins/supplements, stress reduction, mental/cognitive activities.**

|  |  | "Elevated" Amyloid (N = 50) | "Not-Elevated" Amyloid (N = 30) |
|---|---|---|---|
|  | **Made No Change** | **11 (22%)** | **10 (33%)** |
|  | **Made Any Change** | **39 (78%)** | **20 (67%)** |
| **Changes by Domain**[a] | **Preventive** | **36 (72%)** | **16 (53%)** |
|  | *Exercise* | 22 | 8 |
|  | *Diet* | 17 | 13 |
|  | *Cognitive Activity* | 23 | 5 |
|  | *Medication/Vitamins/Supplements* | 9 | 0 |
|  | *Sleep* | 3 | 0 |
|  | *Quit/reduce alcohol, smoking, marijuana* | 4 | 0 |
|  | **Quality of Life** | **17 (34%)** | **6 (20%)** |
|  | *Socializing* | 5 | 2 |
|  | *Adopting a Positive Outlook* | 5 | 1 |
|  | *Practicing Religion/Spirituality* | 3 | 1 |
|  | *Volunteering* | 4 | 0 |
|  | *Meditating* | 8 | 1 |
|  | *Playing Music* | 2 | 0 |
|  | *Adopting a Pet* | 2 | 2 |
|  | **Other** | **18 (36%)** | **1 (3%)** |
|  | *Reading and Learning about AD Research* | 11 | 1 |
|  | *Adopting Strategies to Compensate* | 9 | 0 |

[a]Codes are not mutually exclusive. Responses were coded for whether a change in each domain was mentioned.

helps reverse some of the effects of Alzheimer's" or "[an article] said that one way that could conceivably reduce your chances of suffering memory problem later on is to keep your brain active."

## Future plans

As shown in Table 4, nearly three-quarters of participants who received an "elevated" amyloid PET scan result (36 of 50 [72%]) described contemplating or making changes to their future plans. Across T1 and T2, the most common changes were in the domains of planning for use of leisure time (19 of 50 [38%]), financial planning (15 [30%]), medical, legal, or general planning (14 [28%]), and adjustment of living arrangements (13 [26%]).

When asked directly, most participants who had received an "elevated" result denied that changes in their future plans were due to their amyloid imaging result. Instead they attributed the changes to their age or life-stage, or to changes in their family situation. For instance, one woman explained that her decision to "clear out the crap" in her house was "[n]ot as a result of the PET scan" but because family members were "all aging, and . . . all in various degrees of decrepitude, so that makes me face my future more." Another reported that "establishing trusts or investments" had "nothing to do with memory. It's just got to [do] with common sense." Several did acknowledge, however, that the "elevated" result might be an extra push to engage in planning. A number discussed making changes to avoid "becom[ing] a burden to any of my family."

More than half of the participants who received a "not-elevated" result (17 of 30 [57%]) reported that they were not contemplating or making changes to their future plans across T1

**Table 4. Future planning: Material derived from responses to the questions: "How did learning your result change how you feel about your future? Are you changing or reassessing any plans in your life since learning the result?".**

| | | "Elevated" Amyloid (N = 50) | "Not-Elevated" Amyloid (N = 30) |
|---|---|---|---|
| | **Made No Change** | **14 (28%)** | **17 (57%)** |
| | **Made Any Change** | **36 (72%)** | **13 (43%)** |
| **Changes by Domain**[a] | **Leisure Time & Activities** *E.g., traveling, checking items off "bucket list," doing enjoyable activities now rather than putting them off for later* | **19 (38%)** | **4 (13%)** |
| | **Financial Planning** *E.g., "getting finances in order," meeting with financial planner, reviewing/ updating accounts and investments, considering/purchasing insurance, spending more, saving more* | **15 (30%)** | **4 (13%)** |
| | **Medico-Legal Planning** *E.g., "getting affairs in order," meeting with lawyer, reviewing/updating will, other estate planning, power of attorney, conversations with loved ones about medical and end of life wishes* | **14 (28%)** | **3 (10%)** |
| | **Living Arrangements** *E.g., downsizing or selling properties, considering long-term care facilities or continuing care retirement communities, moving closer to or in with family, home repairs/ renovations to accommodate aging and changing abilities, organizing/decluttering* | **13 (26%)** | **5 (17%)** |
| | **Employment** *E.g., balancing work and leisure, retiring, reducing workload, considering when/if to retire, switching careers, considering whether to take new job* | **6 (12%)** | **1 (3%)** |
| | **Activities of Daily Living** *E.g., considering/making plans related to potential changes in capabilities such as home maintenance, cleaning, and yard work, taking care of loved ones, driving* | **4 (8%)** | **0 (0%)** |

[a]Codes are not mutually exclusive. Responses were coded for whether a change in each domain was mentioned.

and T2. Some explained that, after getting the "not-elevated" result, "[f]or better or worse, I don't feel as compelled to make long term plans." Participants were asked to consider the counterfactual—that is, what they would have done had their amyloid imaging result been "elevated." Most (86% across T1 and T2) stated that they would have changed their future plans. Their hypothesized changes were consistent with the actual changes that participants who received an "elevated result" reported contemplating or making.

Among participants still working, a third of those who received an "elevated" result (6 of 19 [32%]) reported making or contemplating changes to their employment status, whereas only one individual who received a "not-elevated" result (1 of 10 [10%]) described making or contemplating such a change. Participants with elevated brain amyloid spoke to the potential for changes in job performance due to cognitive decline. One participant explained, "[A] factor. . .that I have in the back of my mind, is whether [the amyloid] will affect my teaching ability, whether I should do that [continue teaching] or not." Participants with elevated brain amyloid also perceived tradeoffs between work and leisure. One participant explained, "Now I'm thinking, 'Oh, gosh. Maybe I should cut back on my working. Maybe I should live life now while I have a chance and spend all my retirement money traveling among other things.' . . . I need to think more carefully about if my time is limited how much time do I want to spend working?"

## Discussion

SOKRATES studied how cognitively unimpaired older adults react to learning the result of a PET scan measuring brain amyloid, an AD biomarker. Though conducted in a research context rather than a clinical one, A4's screening process of assessment and testing, disclosure of test results, and if indicated, provision of therapy closely mirrors clinical practice in the context of other serious diseases. Thus, SOKRATES participants' experiences provide important insights into the future experience of living with a preclinical AD diagnosis. In particular, comparing the results of participants who received an "elevated" amyloid PET scan result to the

results of participants who received a "not-elevated" result informs our understanding of how clinicians should talk about amyloid PET scans with patients in the pre- and post-scan context.

SOKRATES participants generally understood the key point of the A4 amyloid imaging disclosure process: an "elevated" amyloid PET scan result means a person has an increased but presently unquantifiable risk of developing AD dementia.[18] Notably, participants differed in their interest in further details about the amyloid PET scan result. Participants who received an "elevated" result often wanted more information, whereas those who received a "not-elevated" result were generally satisfied with detail they received. Participants with elevated amyloid viewed the result as providing an explanation for perceived memory impairments. In contrast, as seen in prior work, participants who received a "not-elevated" result reinterpreted perceived memory impairments as normal aging.[19]

Consistent with prior work on disclosure of risk for AD, receipt of an "elevated" result sparked negative emotions but did not lead to extreme distress.[20] These negative emotions decreased but did not entirely dissipate with time. Our findings support the emerging consensus on the safety of disclosing amyloid imaging results to cognitively unimpaired persons following pre-test assessments of knowledge and psychological well-being.[21–24] Further, they are consistent with studies concluding that it is safe to disclose amyloid imaging results to adults with MCI and dementia.[25,26]

Post-disclosure, participants who received an "elevated" result reported contemplating and making changes to health behaviors and future plans to a greater extent than participants who received a "not-elevated" results. Participants with elevated brain amyloid did not uniformly ascribe these changes to their amyloid PET scan result. Yet, given both the relatively higher frequency of changes reported by those with elevated brain amyloid and the responses of participants who did not have elevated brain amyloid to a counterfactual question about what changes they would make had their result been "elevated," we infer that the "elevated" amyloid imaging result was a key driver of change. Further supporting this inference, our findings are consistent with changes seen in APOE ε4 carriers who receive information about their risk of AD dementia.[27,28] Of particular interest, we found that disclosure of an "elevated" amyloid PET scan result brings into sharp relief tradeoffs related to time and money. Participants reflected on spending money for pleasure versus saving money in anticipation of future care expenses and, similarly, on working to save money versus retiring to enjoy life while still cognitively unimpaired.

Participants with elevated brain amyloid viewed the amyloid PET scan result as a serious, sensitive piece of health information. They highlighted its unique implications for identity, self-determination, and social interactions. Public stigma of AD—the attitudes and beliefs of the general public towards persons with AD—and self-stigma—which occurs when people internalize negative public attitudes—provide important context to our findings.[29,30] Stigma was reflected in participants' circumspect approaches to sharing an "elevated" amyloid PET scan result with others and also their concerns about how the "elevated" result would change how they are perceived and treated by others. Further research is needed to understand how advances in AD diagnosis, testing, and treatment may alter the experience of AD stigma. Participants with elevated brain amyloid expressed concerns about discrimination in the contexts of employment and insurance that highlight the limited ability of current laws and policies to protect those with preclinical AD. The Genetic Information Nondiscrimination Act, for example, does not provide meaningful protections against discrimination based on AD biomarkers for Americans.[31] Advances in the diagnosis and treatment of AD will need to be matched with policy innovations to protect proactive patients.

Finally, irrespective of their brain amyloid status, SOKRATES participants were mindful that their amyloid PET scan result had implications for themselves and also for others.

Participants with elevated brain amyloid expressed concern that they might burden their families if they developed AD, while others reported changing their living situation—for example, moving closer to an adult child—to facilitate future caregiving. More than 16 million adults across the United States currently provide informal dementia care and serve as the backbone of the nation's long-term care system for persons with dementia.[32,33] Future studies should examine the effects of biomarker disclosure on study partners or "pre-caregivers" asked to monitor the cognition, function, and well-being of individuals with preclinical AD.[25,34] Monitoring a cognitively unimpaired adult with elevated biomarkers is different work than being a caregiver for a person with dementia. Pre-caregivers' reactions to biomarker results will likely differ from caregivers' reactions because pre-caregivers are not performing the physical and emotional labor of caregiving, but instead, anticipating it.

## Limitations

This is a small sample, and SOKRATES participants—though reflective of A4 and LEARN participants—are homogeneous. Further, SOKRATES participants all passed assessments of psychological well-being and agreed to learn their amyloid PET scan result in order to determine eligibility to participate in A4. Individuals who might have had adverse responses to the information or who were not interested in enrolling in A4 were excluded. This limits generalizability. SOKRATES was a qualitative study and therefore did not measure quantitative psychometrics for comparisons of mood pre- and post-disclosure.[25] The A4 Study Team did gather psychological measures, which are reported elsewhere.[35] All SOKRATES participants underwent a standardized disclosure process; while that is a strength of the present study, we note that our findings may be contingent on the disclosure process. There was no pre-disclosure interview to understand baseline self-perceptions or expectations about the amyloid PET scan result. This is a direction for future research. We suggest additional research to examine AD biomarker disclosure across a larger, more diverse sample (e.g., age, race/ethnicity, family history of AD) to more fully understand the experience of receiving an AD biomarker result and how those experiences might vary across groups.

## Conclusion

These findings demonstrate the need for additional research on the effect on patients and their families of learning information about preclinical AD to accompany along with research to understand the pathophysiological changes occurring in the brain.

## Supporting information

**S1 Interview guide.**
(DOCX)

**S2 Interview guide.**
(DOCX)

## Author Contributions

**Conceptualization:** Emily A. Largent, Kristin Harkins, Pamela Sankar, Jason Karlawish.

**Formal analysis:** Emily A. Largent, Kristin Harkins, Sara Hachey, Pamela Sankar, Jason Karlawish.

**Funding acquisition:** Jason Karlawish.

**Investigation:** Kristin Harkins, Christopher H. van Dyck, Jason Karlawish.

**Project administration:** Kristin Harkins.

**Supervision:** Emily A. Largent, Kristin Harkins.

**Writing – original draft:** Emily A. Largent, Kristin Harkins, Jason Karlawish.

**Writing – review & editing:** Emily A. Largent, Kristin Harkins, Christopher H. van Dyck, Sara Hachey, Pamela Sankar, Jason Karlawish.

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
