## [Decision Letter · Decision Letter 0]

13 Nov 2019

PONE-D-19-22767

Cognitively Unimpaired Adults’ Reactions to Amyloid Disclosure

PLOS ONE

Dear Dr. Largent,

Thank you for submitting your manuscript to PLOS ONE. After careful consideration, we feel that it has merit but does not fully meet PLOS ONE’s publication criteria as it currently stands. Therefore, we invite you to submit a revised version of the manuscript that addresses the points raised during the review process.

Please consider all of the reviewer's suggestions, particularly those that advise adding more details about the methods and renaming groups. Also, please assure that you have referenced all pertinent literature on the topic (the reviewers provide some citations). 

We would appreciate receiving your revised manuscript by Dec 28 2019 11:59PM. To enhance the reproducibility of your results, we recommend that if applicable you deposit your laboratory protocols in protocols.io, where a protocol can be assigned its own identifier (DOI) such that it can be cited independently in the future. For instructions see: http://journals.plos.org/plosone/s/submission-guidelines#loc-laboratory-protocols

We look forward to receiving your revised manuscript.

Kind regards,

Erin Bouldin, MPH, PhD

Academic Editor

PLOS ONE

Journal Requirements:

1. Please include a copy of the topic/interview guide used in the study, in both the original language and English, as Supporting Information, or include a citation if it has been published previously.

2. Please provide additional details regarding participant consent. In the Methods section, please state why it was not possible to obtain written consent, how verbal consent was recorded and whether the ethics committee approved this consent procedure. If your study included minors, state whether you obtained consent from parents or guardians.

Reviewers' comments:

Reviewer's Responses to Questions

**Comments to the Author**

1. Is the manuscript technically sound, and do the data support the conclusions?

Reviewer #1: Partly

Reviewer #2: Yes

2. Has the statistical analysis been performed appropriately and rigorously? 

Reviewer #1: No

Reviewer #2: N/A

3. Have the authors made all data underlying the findings in their manuscript fully available?

Reviewer #1: No

Reviewer #2: No

4. Is the manuscript presented in an intelligible fashion and written in standard English?

Reviewer #1: No

Reviewer #2: Yes

5. Review Comments to the Author

Reviewer #1: Submitted article does not identify/clarify whether or not intended for special issue on "reward and decision making", so general submission to PLOS ONE assumed without consideration for special issue.

Title: "Cognitively Unimpaired Adults’ Reactions to Amyloid Disclosure"

There is a problem with use of the phrase "amyloid disclosure" in the title. Why not use the full phrase "amyloid imaging results disclosure"? What is the meaning of the shortened phrase "amyloid disclosure"? Does it refer to which testing or imaging of which kind of amyloid? Is it possible to "disclose amyloid"? Or is that a non-sensical phrase for which example precedents do not exist in published medical science? Certainly, we do not publish medical scientific reports on "blood disclosure" or "bone disclosure" or any other kind of bio-tissue or bio-molecule!!!

Abstract contains other shortened phrases such as "elevated participants" and "not-elevated participants" that also result in potentially confusing and/or non-sensical abuses of English language. Again, this reviewer objects to these misuses of the English language. Presumably, the research participants are neither elevated nor not-elevated! Rather, their amyloid imaging results have been interpreted as elevated or not-elevated.

Abstract contains the phrase "urgent need to understand the effects of biomarker disclosure on this population" which raises the question: what is the "urgency"? In clinical medical contexts, use of the words "emergent" or "urgent" do have relevant and important meanings. However, majority of evidence published so far on amyloid imaging results disclosure suggests that the risks if any are minimal, and certainly do not rise to the level or either emergent or urgent. So what is the rationale or justification for use of the word "urgent"? Why over-dramatize and raise an alarm with the word "urgent" if minimal if any harm will occur? And if the authors do believe that harm will occur then they should provide the explanation, rationale and evidence for the harm that they fear will occur with probability estimates for the numbers of potential patients impacted. How do these predicted numbers compare to the numbers of persons involved in the current opiates drug overdose epidemic plaguing America today? And what constitutes an emergency or urgency? The authors should be careful before using language that creates drama frightening people unnecessarily.

Data availability reported as "fully available without restriction" with additional statement that "All relevant data are within the manuscript and its Supporting Information files." However, this reviewer did not find any relevant raw data in either the manuscript or supporting information. There are no tables or URL links to repositories of raw data found anywhere in the manuscript or supporting information files. The manuscript contains only tables of analyzed results.

Introduction contains the statement "The extant empirical literature on such disclosure is extremely limited.(11),(12)" However, this statement is neither correct nor current and up-to-date. Despite presence of bibliography with a total of 30 references, some important literature has been omitted. If the total number of references must be constrained, then would not some of these missing references be more important than some of those that were mentioned? Criteria for inclusion of literature citations should be relevance to discussion and argument of issues and claims in the submitted manuscript.

Missing literature:

https://www.alz.org/aaic/program/education-workshop-neuroimaging.asp

Book chapter associated with talk and slide presentation at AAIC 2019 Los Angeles on Fri Jul 12 at 11:30 AM

"Disclosure of Amyloid Imaging Results" 2019 by Jennifer H. Lingler.

This work important as a recent comprehensive survey that should be up to date for amyloid imaging results disclosure studies as of the AAIC conference in July 2019; it appears that none of the papers by Lingler, neither recent nor past, have been discussed by the authors of the manuscript submitted to PLOS ONE.

http://mhfmjournal.com/pdf/MHFM-120.pdf

"Safety of Disclosing Amyloid Imaging Results to MCI and AD Patients" 2018 Taswell et al

This work important as the largest size study to date of amyloid imaging results disclosure, also important for the careful use of robust statistical methods in the data analysis. This important clinical trial has not been cited and discussed by the authors of the manuscript submitted to PLOS ONE.

Methods contains the statement "Psychological measures of well-being are collected for all A4 and LEARN participants and are not reported here." Why not?!? What is the rationale and justification for NOT reporting and analyzing the results from the psychological measures of well-being for the research participants? If the data are available and analyzed correctly, then they should be reported.

Results contain the statement "No notable differences in results were observed by age or gender." However, there are no group-wise results based on age group based analysis presented in the manuscript with evidence to support this claim which contradicts intuition and common life experience. The authors should either present the data and results to substantiate their claims, or at least, offer a hypothetical explanation for their failure to find differences between age groups. What about younger age groups in their 50s, 40s or 30s? This reviewer remains skeptical that there would be no differences between age groups. It's just a matter of clarifying the specific ages of the age groups over the life span.

Reviewer #2: Largent et al performed qualitatitive research to assess the reaction to a positive/negative amyloid PET result in cognitively normal individuals. This is an important topic as increasing numbers of cognitively healthy persons are undergoing amyloid PET for research and the return of study results is very much wanted by study participants a priori.

The number of participants is high, with a high proportion of persons (50 out of 80) with an elevated amyloid PET result. Overall the study appears to confirm that knowledge of an elevated amyloid PET may have negative consequences, mainly because of stigma associated with the disease.

Major comments:

1. On several occasions (e.g. p 6) the authors allude to the fact that the participants perceived memory problems beforehand. It would be of interest to better describe the cognitive complaints/status of the participants prior to disclosure: what was their motivation for enrolling in the study, did they fulfill criteria for subjective cognitive decline, was there a difference in test scores between those who had a positive versus a negative amyloid PET? If a person has subjective memory complaints, the elevated amyloid PET result may have a very different meaning than when no complaints existed beforehand and cognition was intact. The authors acknowledge in the Limitations section that an interview prior to disclosure could have resolved some of these questions.

Minor comments

1. The disclosure process is described in a previous paper. This essential information may be summarized in a few sentences also in the current paper: which information was disclosed, e.g. in terms of positive and negative predictive value. By whom (physician, same researcher across all participants etc). Did the authors also use written information prior to or during the disclosure? It may be worthwile mentioning in the discussion that some of the reactions may be contingent on these factors.

2. I suggest the authors find a better term for ‘elevated participants’.

3. An analogous qualitative research study was performed in amnestic MCI and disclosure of amyloid PET in a research context by Vanderschaeghe et al. Since the research method is similar and several elements in the participants’ reactions are similar, the authors may consider to refer to this study.

6. PLOS authors have the option to publish the peer review history of their article (what does this mean?). If published, this will include your full peer review and any attached files.

Reviewer #1: No

Reviewer #2: No

---

## [Author Response · Author response to Decision Letter 0]

18 Nov 2019

Reviewer #1: 

Submitted article does not identify/clarify whether or not intended for special issue on "reward and decision making", so general submission to PLOS ONE assumed without consideration for special issue.

• We apologize for any confusion; the cover letter noted that this paper was submitted to the “Early Diagnosis and Treatment of Alzheimer’s Disease” Call for Papers. 

Title: "Cognitively Unimpaired Adults’ Reactions to Amyloid Disclosure." There is a problem with use of the phrase "amyloid disclosure" in the title. Why not use the full phrase "amyloid imaging results disclosure"? What is the meaning of the shortened phrase "amyloid disclosure"? Does it refer to which testing or imaging of which kind of amyloid? Is it possible to "disclose amyloid"? Or is that a non-sensical phrase for which example precedents do not exist in published medical science? Certainly, we do not publish medical scientific reports on "blood disclosure" or "bone disclosure" or any other kind of bio-tissue or bio-molecule!!!

• We have changed the title to: Cognitively Unimpaired Adults’ Reactions to Disclosure of Amyloid PET Scan Results. 

Abstract contains other shortened phrases such as "elevated participants" and "not-elevated participants" that also result in potentially confusing and/or non-sensical abuses of English language. Again, this reviewer objects to these misuses of the English language. Presumably, the research participants are neither elevated nor not-elevated! Rather, their amyloid imaging results have been interpreted as elevated or not-elevated.

• We note that Reviewer 2 also shared this concern about use of “elevated” and “not-elevated.” We have made responsive edits throughout the manuscript to reflect that we are discussing “elevated” and “not-elevated” amyloid PET scan result. 

Abstract contains the phrase "urgent need to understand the effects of biomarker disclosure on this population" which raises the question: what is the "urgency"? In clinical medical contexts, use of the words "emergent" or "urgent" do have relevant and important meanings. However, majority of evidence published so far on amyloid imaging results disclosure suggests that the risks if any are minimal, and certainly do not rise to the level or either emergent or urgent. So what is the rationale or justification for use of the word "urgent"? Why over-dramatize and raise an alarm with the word "urgent" if minimal if any harm will occur? And if the authors do believe that harm will occur then they should provide the explanation, rationale and evidence for the harm that they fear will occur with probability estimates for the numbers of potential patients impacted. How do these predicted numbers compare to the numbers of persons involved in the current opiates drug overdose epidemic plaguing America today? And what constitutes an emergency or urgency? The authors should be careful before using language that creates drama frightening people unnecessarily.

• We have deleted the word “urgent” from the abstract as it was not our intention to unnecessarily frighten anyone. 

Data availability reported as "fully available without restriction" with additional statement that "All relevant data are within the manuscript and its Supporting Information files." However, this reviewer did not find any relevant raw data in either the manuscript or supporting information. There are no tables or URL links to repositories of raw data found anywhere in the manuscript or supporting information files. The manuscript contains only tables of analyzed results.

• This was in error. The transcripts are not fully available without restriction as they contain identifiable information. 

Introduction contains the statement "The extant empirical literature on such disclosure is extremely limited.(11),(12)" However, this statement is neither correct nor current and up-to-date. Despite presence of bibliography with a total of 30 references, some important literature has been omitted. If the total number of references must be constrained, then would not some of these missing references be more important than some of those that were mentioned? Criteria for inclusion of literature citations should be relevance to discussion and argument of issues and claims in the submitted manuscript.

• Reviewer 1’s point that there is a literature on AD biomarker disclosure is well taken. We have revised the sentence as follows: “The extant empirical literature on disclosure of amyloid PET scan results to cognitive unimpaired adults is, however, limited.” 

Missing literature: 

Book chapter associated with talk and slide presentation at AAIC 2019 Los Angeles on Fri Jul 12 at 11:30 AM. "Disclosure of Amyloid Imaging Results" 2019 by Jennifer H. Lingler (https://www.alz.org/aaic/program/education-workshop-neuroimaging.asp). This work important as a recent comprehensive survey that should be up to date for amyloid imaging results disclosure studies as of the AAIC conference in July 2019; it appears that none of the papers by Lingler, neither recent nor past, have been discussed by the authors of the manuscript submitted to PLOS ONE.

• We have added a citation to Lingler & Klunk (2013) as that paper specifically recommends against disclosure of amyloid imaging results to cognitively normal research participants. 

• We have also added a citation to Kim and Lingler (2019). 

"Safety of Disclosing Amyloid Imaging Results to MCI and AD Patients" 2018 Taswell et al (http://mhfmjournal.com/pdf/MHFM-120.pdf). This work important as the largest size study to date of amyloid imaging results disclosure, also important for the careful use of robust statistical methods in the data analysis. This important clinical trial has not been cited and discussed by the authors of the manuscript submitted to PLOS ONE.

• We have added a citation to Taswell et al (2018). We note, however, that this study looked at disclosure of results to adults with MCI and dementia rather than to cognitively unimpaired adults. 

Methods contains the statement "Psychological measures of well-being are collected for all A4 and LEARN participants and are not reported here." Why not?!? What is the rationale and justification for NOT reporting and analyzing the results from the psychological measures of well-being for the research participants? If the data are available and analyzed correctly, then they should be reported.

• The psychological measures are proprietary to the A4 and LEARN studies and are not presently available for analysis or for publication by the SOKRATES team. We have deleted this sentence as unnecessary. 

Results contain the statement "No notable differences in results were observed by age or gender." However, there are no group-wise results based on age group based analysis presented in the manuscript with evidence to support this claim which contradicts intuition and common life experience. The authors should either present the data and results to substantiate their claims, or at least, offer a hypothetical explanation for their failure to find differences between age groups. What about younger age groups in their 50s, 40s or 30s? This reviewer remains skeptical that there would be no differences between age groups. It's just a matter of clarifying the specific ages of the age groups over the life span.

• We have added a sentence explaining that Analyses included searching for differences among groups based on demographic variables.

• SOKRATES enrolled only participants aged 65 to 85 – because one eligibility criterion for the parent A4 and LEARN studies was being 65 to 85 years old. In our analyses, we looked at two age groups: 65 to 74 and 75+. We did not find differences by age between these two groups. 

• It is an open question whether there would be differences by age if younger individuals received amyloid PET scan results. As that is not a question that can be answered with our present data set, we now note this as a direction for future research. 

Reviewer #2: 

Largent et al performed qualitative research to assess the reaction to a positive/negative amyloid PET result in cognitively normal individuals. This is an important topic as increasing numbers of cognitively healthy persons are undergoing amyloid PET for research and the return of study results is very much wanted by study participants a priori.

• We are glad that Reviewer 2 shares our belief that this is an important and timely topic for research and also the future of clinical care. 

The number of participants is high, with a high proportion of persons (50 out of 80) with an elevated amyloid PET result. Overall the study appears to confirm that knowledge of an elevated amyloid PET may have negative consequences, mainly because of stigma associated with the disease.

• Thank you – we believe that the large sample size is a strength of our study.

On several occasions (e.g. p 6) the authors allude to the fact that the participants perceived memory problems beforehand. It would be of interest to better describe the cognitive complaints/status of the participants prior to disclosure: what was their motivation for enrolling in the study, did they fulfill criteria for subjective cognitive decline, was there a difference in test scores between those who had a positive versus a negative amyloid PET? If a person has subjective memory complaints, the elevated amyloid PET result may have a very different meaning than when no complaints existed beforehand and cognition was intact. The authors acknowledge in the Limitations section that an interview prior to disclosure could have resolved some of these questions.

• We have added additional information in the methods section to note that inclusion criteria for A4/LEARN required that participants were cognitively unimpaired. They scored within normal limits on baseline cognitive testing and were rated a zero on the CDR. Subjective cognitive complaints were permitted. 

• Cognitive measures were collected in the A4 and LEARN studies but are not presently available for analysis or for publication by the SOKRATES team. The SOKRATES team did not collect cognitive measures. We agree with Reviewer 2 that this is a limitation and have noted it as such in our limitations section. 

The disclosure process is described in a previous paper. This essential information may be summarized in a few sentences also in the current paper: which information was disclosed, e.g. in terms of positive and negative predictive value. By whom (physician, same researcher across all participants etc). Did the authors also use written information prior to or during the disclosure? It may be worthwile mentioning in the discussion that some of the reactions may be contingent on these factors.

• We thank Reviewer 2 for this suggestion and have added additional detail about the disclosure process on P4 of the manuscript. 

• On page 17, we note that some of the reactions may be contingent on the standardized disclosure process. 

I suggest the authors find a better term for ‘elevated participants’.

• We note that Reviewer 1 also shared this concern. We have made responsive edits throughout the manuscript to reflect that we are discussing “elevated” and “not-elevated” amyloid PET scan result. 

An analogous qualitative research study was performed in amnestic MCI and disclosure of amyloid PET in a research context by Vanderschaeghe et al. Since the research method is similar and several elements in the participants’ reactions are similar, the authors may consider to refer to this study.

• We have added a citation to Vanderschaeghe et al (2017). 

Other: 

Please include a copy of the topic/interview guide used in the study, in both the original language and English, as Supporting Information, or include a citation if it has been published previously.

• As requested, we are including a copy of each interview guide used in the study. 

Please provide additional details regarding participant consent. In the Methods section, please state why it was not possible to obtain written consent, how verbal consent was recorded and whether the ethics committee approved this consent procedure. If your study included minors, state whether you obtained consent from parents or guardians.

• We have updated the Methods section (PP 5-6) to provide additional details about consent. The IRB approved the waiver of written informed consent because the interviews were telephonic and the study was minimal risk. 

• Our study did not include minors.

---

## [Decision Letter · Decision Letter 1]

28 Jan 2020

PONE-D-19-22767R1

Cognitively Unimpaired Adults’ Reactions to Disclosure of Amyloid PET Scan Results 

PLOS ONE

Dear Dr. Largent,

Thank you for submitting your manuscript to PLOS ONE. After careful consideration, we feel that it has merit but does not fully meet PLOS ONE’s publication criteria as it currently stands. Therefore, we invite you to submit a revised version of the manuscript that addresses the points raised during the review process.

Please address Reviewer 1's remaining concern about the limitations as they relate to psychometric measures. While I appreciate Reviewer 1's concern about data availability, the current statement you have provided, which focuses on the potential for identification of participants, aligns with PLOS ONE's data availability policy so does not need to be changed.

We would appreciate receiving your revised manuscript by Mar 13 2020 11:59PM. To enhance the reproducibility of your results, we recommend that if applicable you deposit your laboratory protocols in protocols.io, where a protocol can be assigned its own identifier (DOI) such that it can be cited independently in the future. For instructions see: http://journals.plos.org/plosone/s/submission-guidelines#loc-laboratory-protocols

We look forward to receiving your revised manuscript.

Kind regards,

Erin Bouldin, MPH, PhD

Academic Editor

PLOS ONE

Reviewers' comments:

Reviewer's Responses to Questions

**Comments to the Author**

1. If the authors have adequately addressed your comments raised in a previous round of review and you feel that this manuscript is now acceptable for publication, you may indicate that here to bypass the “Comments to the Author” section, enter your conflict of interest statement in the “Confidential to Editor” section, and submit your "Accept" recommendation.

Reviewer #1: (No Response)

Reviewer #2: All comments have been addressed

2. Is the manuscript technically sound, and do the data support the conclusions?

Reviewer #1: Partly

Reviewer #2: Yes

3. Has the statistical analysis been performed appropriately and rigorously? 

Reviewer #1: No

Reviewer #2: Yes

4. Have the authors made all data underlying the findings in their manuscript fully available?

Reviewer #1: No

Reviewer #2: Yes

5. Is the manuscript presented in an intelligible fashion and written in standard English?

Reviewer #1: Yes

Reviewer #2: Yes

6. Review Comments to the Author

Reviewer #1: Data availability: Authors state that "The transcripts are not fully available without restriction as they contain identifiable information". However, data in the form of interview transcripts could be made available simply by redacting the "identifiable information" and de-identifying the raw source material of the interview transcripts. De-identifying clinical research data should be considered a standard expected and required step in the process of conducting clinical trials and publishing the results in a journal that expects the data to be made publicly available. If the authors wish to argue that the data cannot be shared publicly because they contain identifying information about participants, then perhaps they should have considered submission to a different journal that does not require publishing the data. Otherwise, the authors should consider conducting clinical trials with a protocol and methods that plan in advance appropriately to de-identify and anonymize the clinical research data. Did the original informed consent from participants request their agreement to publishing de-identified data from the clinical research trial?

Limitations: Authors have not discussed, but should discuss, a major limitation of their study. They did not perform or report any quantitative psychometrics for comparisons of mood with after-disclosure versus before-disclosure comparisons evaluating individual change scores for participants as reported by Taswell et al in 2018 MHFM.

Authors also recommend that "Future studies should examine the effects of biomarker disclosure on study partners or pre-caregivers asked to monitor the cognition, function, and well-being of individuals with preclinical AD". But they did not cite the first publication of this general recommendation by Taswell et al in 2018 MHFM as follows: "we believe that a more productive area of possible future research would be evaluation of mood scale psychometrics for the patient’s primary caregiver instead of psychometrics for the patient". They should cite Taswell et al 2018 MHFM for the recommendation, and then discuss what they think might be differences between cognitively impaired patients and cognitively non-impaired volunteers and their respective families and study partner / caregivers.

Reviewer #2: (No Response)

7. PLOS authors have the option to publish the peer review history of their article (what does this mean?). If published, this will include your full peer review and any attached files.

Reviewer #1: No

Reviewer #2: Yes: Rik Vandenberghe

---

## [Author Response · Author response to Decision Letter 1]

30 Jan 2020

Reviewer #1: 

Limitations: Authors have not discussed, but should discuss, a major limitation of their study. They did not perform or report any quantitative psychometrics for comparisons of mood with after-disclosure versus before-disclosure comparisons evaluating individual change scores for participants as reported by Taswell et al in 2018 MHFM.

• In the limitations section, we now note that because SOKRATES is a qualitative study, we did not collect quantitative psychometrics. Such measures were, however, collected as part of A4, and will be published elsewhere (manuscript under review). 

• Additionally, we have included a citation to Taswell et al (2018) here (Reference 25). 

Authors also recommend that "Future studies should examine the effects of biomarker disclosure on study partners or pre-caregivers asked to monitor the cognition, function, and well-being of individuals with preclinical AD". But they did not cite the first publication of this general recommendation by Taswell et al in 2018 MHFM as follows: "we believe that a more productive area of possible future research would be evaluation of mood scale psychometrics for the patient’s primary caregiver instead of psychometrics for the patient". They should cite Taswell et al 2018 MHFM for the recommendation, and then discuss what they think might be differences between cognitively impaired patients and cognitively non-impaired volunteers and their respective families and study partner / caregivers.

• We have now noted that “Monitoring a cognitively unimpaired adult with elevated biomarkers is different work than being a caregiver for a person with dementia. Pre-caregivers reactions to biomarker results will likely differ from caregivers’ reactions because pre-caregivers are not performing the physical and emotional labor of caregiving, but instead, anticipating it.”

• Additionally, we have included a citation to Taswell et al (2018) here (Reference 25). 

Reviewer #2: 

(No Response)

---

## [Editor Report · Decision Letter 2]

31 Jan 2020

Cognitively Unimpaired Adults’ Reactions to Disclosure of Amyloid PET Scan Results

PONE-D-19-22767R2

Dear Dr. Largent,

We are pleased to inform you that your manuscript has been judged scientifically suitable for publication and will be formally accepted for publication once it complies with all outstanding technical requirements.

With kind regards,

Erin Bouldin, MPH, PhD

Academic Editor

PLOS ONE

---

## [Editor Report · Acceptance letter]

4 Feb 2020

PONE-D-19-22767R2 

Cognitively Unimpaired Adults’ Reactions to Disclosure of Amyloid PET Scan Results 

Dear Dr. Largent:

I am pleased to inform you that your manuscript has been deemed suitable for publication in PLOS ONE. Congratulations! Your manuscript is now with our production department. 

With kind regards,

on behalf of

Dr. Erin Bouldin 

Academic Editor

PLOS ONE